# Increase of Circulating Endothelial Progenitor Cells and Released Angiogenic Factors in Children with Moyamoya Arteriopathy

**DOI:** 10.3390/ijms24021233

**Published:** 2023-01-08

**Authors:** Gemma Gorla, Tatiana Carrozzini, Giuliana Pollaci, Antonella Potenza, Sara Nava, Francesco Acerbi, Paolo Ferroli, Silvia Esposito, Veronica Saletti, Emilio Ciusani, Aida Zulueta, Eugenio A. Parati, Anna Bersano, Laura Gatti, Ignazio G. Vetrano

**Affiliations:** 1Laboratory of Neurobiology and UCV, Neurology IX Unit, Fondazione IRCCS Istituto Neurologico Carlo Besta, 20133 Milan, Italy; 2Department of Pharmacological and Biomolecular Sciences, Università di Milano, 20122 Milan, Italy; 3Unità Produzione Terapie Cellulari (UPTC), Fondazione IRCCS Istituto Neurologico Carlo Besta, 20133 Milan, Italy; 4Department of Neurosurgery, Fondazione IRCCS Istituto Neurologico Carlo Besta, 20133 Milan, Italy; 5Experimental Microsurgical Laboratory, Fondazione IRCCS Istituto Neurologico Carlo Besta, 20133 Milan, Italy; 6Developmental Neurology Unit, Fondazione IRCCS Istituto Neurologico Carlo Besta, 20133 Milan, Italy; 7Department of Diagnostic and Technology, Fondazione IRCCS Istituto Neurologico Carlo Besta, 20133 Milan, Italy; 8Istituti Clinici Scientifici Maugeri IRCCS, Neurorehabilitation Unit of Milan Institute, 20138 Milan, Italy; 9Department of Biomedical Sciences for Health, Università di Milano, 20122 Milan, Italy

**Keywords:** pediatric moyamoya, biomarkers, cEPC, ANG-2, VEFG-A, cerebrospinal fluid, plasma

## Abstract

Moyamoya arteriopathy (MMA) is a rare cerebrovascular disorder that causes recurrent ischemic and hemorrhagic strokes, leading young patients to severe neurological deficits. The pathogenesis of MMA is still unknown. The disease onset in a wide number of pediatric cases raises the question of the role of genetic factors in the disease’s pathogenesis. In these patients, MMA’s clinical course, or progression, is largely unclear. By performing a comprehensive molecular and cellular profile in the plasma and CSF, respectively, of MMA pediatric patients, our study is aimed at assessing the levels of circulating endothelial progenitor cells (cEPC) and the release of selected proteins at an early disease stage to clarify MMA pathogenesis and progression. We employed cytofluorimetric methods and immunoassays in pediatric MMA patients and matched control subjects by age and sex. We detected increased levels of cEPC in peripheral blood and an upregulation of angiogenic markers in CSF (i.e., angiopoietin-2 and VEGF-A). This finding is probably associated with deregulated angiogenesis, as stated by the moderate severity of collateral vessel network development (Suzuki III-IV). The absence of significant modulation of neurofilament light in CSF led us to rule out the presence of substantial neuronal injury in MMA children. Despite the limited cohort of pediatric patients, we found some peculiar cellular and molecular characteristics in their blood and CSF samples. Our findings may be confirmed by wider and perspective studies to identify predictive or prognostic circulating biomarkers and potential therapeutic targets for personalized care of MMA pediatric patients.

## 1. Introduction

Moyamoya arteriopathy (MMA) is a rare cerebrovascular disorder characterized by a progressive steno-occlusive lesion of the terminal part of the internal carotid arteries (ICAs) and an association with the development of compensatory unstable collateral vessels, the so-called *Moyamoya vessels* [1,2]. These vascular hallmarks cause recurrent ischemic and hemorrhagic strokes, leading affected patients, often young adults and children, to neurological morbidity and also mortality [3,4,5,6,7]. MMA is frequent in East Asian countries, whereas it is rarely reported in Caucasian people. The annual incidence of MMA is 0.35 to 0.94 per 100,000 people with a bimodal age distribution: the first peak manifests in children between 5 and 9 years, and the second peak between 45 and 49 years [8,9].

The pathogenesis and etiology of MMA are still unknown, even if the association with genetic disorders, the high familial rate, and the strong linkage with variants of the *Ring Finger Protein 213 (RNF213*)/*Mysterin* coding gene in East Asian patients strengthen the role of genetic factors in MMA pathogenesis [10,11,12,13,14,15,16,17]. Furthermore, several reports have implicated RNF213 as a sensor for mitochondrial dysfunction, hypoxia, and inflammation, and most recently, it has been involved in antimicrobial activity and lipid metabolism [16,18,19,20,21]. The emerging and considerable role of genetic background, although not yet outlined, is sustained by a wide number of pediatric cases.

It is strongly believed that MMA results from a complex mechanism in which acquired infectious, inflammatory, and flow dynamic conditions may trigger the disease in genetically susceptible individuals through angiogenic and vasculogenic pathway abnormalities [2]. Notably, stenotic changes observed in MMA are not characterized by lipid pools, inflammatory cells, or macrophage invasion to the subintimal layer as typically seen in atherosclerosis [10]. Moreover, as luminal stenosis progresses in atherosclerosis, outward vascular remodeling occurs with an increased outer vessel diameter, in contrast to the decreased outer diameter that occurs in MMA [22]. Pediatric MMA patients presenting with headache or aspecific symptoms may be misdiagnosed. It is possible that children’s inability to accurately report their symptoms may hinder a timely diagnosis and lead to an increased probability of completed strokes. Common childhood actions, including crying with hyperventilation, may precipitate a stroke or TIA. Cortical vasculature, fully dilated under chronic ischemia, may constrict in response to a decrease in the partial pressure of carbon dioxide secondary to prolonged hyperventilation, finally resulting in a reduction of cerebral perfusion. However, the pathophysiology of MMA in children is not fully understood so far, and some studies on the pediatric population with MMA are limited by ethical issues and small sample sizes due to the rarity of this disease. Nevertheless, it is known that MMA represents one-fifth of the identifiable cerebral arteriopathies in childhood stroke, and it is also the most common cause of cerebrovascular disease in children in East Asia [23,24]. Unlike adults with MMA, who show both intracranial hemorrhages and ischemic stroke, pediatric MMA patients mainly present transient ischemic attacks (TIA) or arterial ischemic strokes [25]. Although stroke is a leading cause of severe long-term disability and death worldwide, pediatric stroke is a rare condition. A recent study defined MMA as the third-most prevalent underlying etiology in pediatric strokes, accounting for 14% of total cases [26,27]. Lee and colleagues reported 8% of arterial strokes in children with MMA [28]. Medical therapy is still unable to stop the progression of arterial disease, and revascularization surgery represents the most effective treatment in pediatric MMA [29,30].

Due to the current lack of knowledge about pathophysiology, several studies focused on the research of different types of biomarkers (e.g., proteins, circulating cells) in various biological matrices (e.g., peripheral blood, cerebrospinal fluid, plasma, urine) with the final aim to unveil novel, putative, noninvasive biomarkers [2,11,31,32,33,34].

In the last few years, altered levels of circulating endothelial progenitor cells (cEPCs) were detected in the biological fluids of MMA pediatric patients. EPCs play an important role in physiological and pathological neovascularization. Noticeably, in a homogeneous cohort of nonoperated Caucasian adult MMA patients, a marked decrease of cEPCs was evident [34]. However, little is known about their role in the early onset of the disease. Despite conflicting data about the modulation of the number of EPCs in children with MMA, a common agreement concerns their defective function, which has been observed both in vitro and in vivo [33,34,35,36].

Because of the detection of altered levels of EPCs, cytokines, chemokines, and growth factors in MMA patients’ biological fluids, impaired angiogenesis and vasculogenesis have been invoked as potential disease mechanisms [2,11,34]. The vascular endothelial growth factor A (VEGF-A) plays a central role in the vasculogenesis and angiogenesis of new blood vessels, and thus the modulation of VEGF-A could represent the effect of angiogenic stimuli in the injured brain district of MMA patients [37]. The VEGF pathway intersects with many other signaling pathways, such as Angiopoietin/Tie [38]. The secreted glycoprotein angiopoietin-2 (ANG-2) is a proangiogenic factor involved in the formation of new vessels, but it also promotes pathological angiogenesis, vascular permeability, and inflammation. Indeed, ANG-2 was overexpressed in multiple inflammatory diseases and implicated both in the direct control of inflammation-related signaling pathways and in the recruitment of inflammation cells [38]. The upregulation of ANG-2, already depicted in ECs as a vessel-destabilizing cytokine, could therefore lead to increased, but defective, angiogenesis and to a protracted condition of general inflammation [39]. Interestingly, autocrine release of ANG-2 mediates cerebrovascular disintegration in MMA [40], and increased serum ANG-2 levels may contribute to pathological abnormal angiogenesis and/or to the instability of vascular structure and function, thus causing brain hemorrhage in adult MMA patients [41]. Matrix metalloproteinase 9 (MMP-9) has an important role in extracellular matrix remodeling and angiogenesis regulation, and it is the main enzyme in the degradation of extracellular matrix proteins. It has been already reported as an important factor in MMA [42,43,44,45]. Interleukin 6 (IL-6) is a soluble mediator with a pleiotropic effect on inflammation, immune response, and hematopoiesis [46]. IL-6 also issues a warning signal in the event of tissue damage [47]. Previous studies have investigated the expression of IL-6 in MMA [48,49].

Although the pathophysiology of MMA is believed to encompass many potential and different mechanisms, the discovery of disease-related biomarkers may represent an important starting point to improve and widen our diagnostic and prognostic tools with the final aim of better defining the disease severity and the risk of stroke. In this perspective, released proteins and circulating cells in plasma and CSF may provide us with a useful combination of molecular and functional disease markers.

The aim of the present study is to perform a molecular and cellular analysis of specific protein factors and circulating endothelial progenitors in the peripheral blood, plasma, and CSF of children with MMA, with the final goal of clarifying the disease pathogenesis and identifying candidate biomarkers that could serve to monitor disease progression in early stages.

## 2. Results

### 2.1. Recruitment of MMA Pediatric Patients and Unrelated Pediatric Subjects

Among the original cohort of more than 150 patients of the GEN-O-MA project [50], 16 MMA pediatric patients, of whom it was possible to collect whole blood and/or CSF samples, were included in the present study (Table 1). The full study methodology has already been reported elsewhere [50]. The selected patients displayed a mean age of 9.2 years (range 3–16 years) with a 50% ratio of female subjects. Of these patients, 75% presented an ischemic cerebrovascular event (43.8% ischemic stroke and 31.3% transient ischemic attack, TIA), whereas nobody had a hemorrhagic stroke. These clinical features are in line with reported evidence in the literature [25]. Patients with an estimated IV Suzuki grade represented 56.25%, of the study population, and subjects characterized by III Suzuki grade amounted to 43.75%. Of the entire cohort, 87.5% (total number) presented steno-occlusive lesions that involved both hemispheres (bilateral form-stenosis), whereas the remaining 12.5% showed a unilateral lesion. Twelve out of sixteen patients underwent MCA-STA (superficial temporal artery) bypass. Five out of sixteen cases were classified as syndromic with two patients suffering from neurofibromatosis Type I and one each from 21 trisomy, 4q distal trisomy, and one still-undetected syndrome. Fourteen unrelated pediatric subjects with a mean age of 7 years (range 1–14 years) (43% female subjects) were selected as a control group. There was no considerable difference in age or sex between MMA patients and unrelated controls (UNR), including subjects suffering from different diseases not closely related to MMA (e.g., primary epilepsy, in which hemodynamic mechanisms have been excluded; multiple sclerosis; intracranial hypertension). All relevant clinical characteristics of the MMA patients are summarized in Table 1.

### 2.2. Circulating Endothelial Progenitor Cell (cEPC) Levels Are Increased in Peripheral Blood of Children with MMA

In order to understand the role of cEPCs in MMA, we measured the cEPC% level in the peripheral blood of pediatric patients as compared to age- or sex-matched control subjects.

The fluorescence-activated cell sorting (FACS) evaluation of CD45^dim^CD34^+^CD133^+^ mononuclear cells (Figure 1) showed a statistically significant increase of the cEPC% value in the set of MMA pediatric patients in comparison with control subjects (0.0137 ± 0.0078 in UNR and 0.0645 ± 0.0674 in MMA patients, respectively, with *p* value= 0.0129 as shown in Figure 2a).

To understand if the level of cEPC could be affected by the neurosurgical intervention in pediatric MMA patients, we compared the cEPC% value in peripheral blood before and after bypass revascularization. We did not find a statistically significant difference between the two groups of MMA patients (Figure 2b).

### 2.3. Angiopoietin-2 (ANG-2) and VEGF-A Level Are Increased in CSF of MMA Pediatric Patients

The angiogenic or inflammatory pathways were investigated through ELISA analysis conducted on the plasma and CSF collected from MMA pediatric patients by comparing them to UNR subjects. The levels of selected protein factors potentially released in plasma and CSF (Ang-2; VEGF-A; MMP-9; IL-6) were measured (Figure 3).

The analyses performed on proteins secreted in plasma samples did not show any difference between MMA and UNR subjects (Figure 3a,b). An analogous analysis in CSF detected a statistically significantly higher level of ANG-2 and VEGF-A in MMA pediatric patients in comparison with UNR subjects (Figure 3c,d), whereas the expression of MMP-9 and IL-6 did not change between the two considered groups of subjects (Figure 3e,f).

### 2.4. Neurofilament Light (NfL) and Glial Fibrillary Acidic Protein (GFAP) Release in Plasma and CSF of MMA Pediatric Patients

Because NfL is considered a crucial biomarker of hypoxic–ischemic brain and a sensitive indicator of white matter axonal damage injury and GFAP release is suggestive of astrocyte activation response to brain injury or neuroperturbative conditions, their levels were measured in plasma and CSF samples from MMA pediatric patients. The quantitative determination of NfL and GFAP with an ultrasensitive and selective digital immunoassay did not show any statistical difference in MMA plasma and CSF samples as compared to UNR subjects (Figure 4). Although not statistically significant, a slight increase of NfL levels was observed in the CSF of MMA pediatric patients (Figure 4c).

## 3. Discussion

The present study attempts to fill the current lack of knowledge about MMA pathophysiology through the identification of a comprehensive molecular and cellular profile of the plasma and CSF of children with MMA. The final purpose is to define novel, putative, noninvasive biomarkers that may be exploited to clarify the disease’s pathogenesis and, from a translational point of view, may serve as potential therapeutic targets. The sixteen pediatric European patients enrolled in the GEN-O-MA project displayed a mean age of 9.2 years (range 3–16 years). As expected, none of them were characterized by hemorrhagic stroke, whereas the majority of patients presented ischemic events encompassing ischemic strokes and transient ischemic attacks, likewise with the common clinical features of MMA children reported elsewhere [27,28,29].

The whole cohort of patients studied presented a III and IV Suzuki grade, suggestive of progressive stenosis of the ICA, or severity of the moyamoya vessel network, and development of collaterals from the ECA (external carotid artery), respectively. All patients were symptomatic for ischemic cerebrovascular events. No children in our series were diagnosed as being in an early or later neuroradiological disease stage (I-II or V-VI). In most cases at our institution, the diagnosis of symptomatic MMA in pediatric population leads to early surgical treatment, thus explaining some specific findings in the current cohort of patients. Specifically, the entire cohort of MMA children presented steno-occlusive lesions involving both hemispheres, leading to a well-established diagnosis of MMA. We did not find any prevalence in the percentage of boys and girls in MMA pediatric patients. This finding differed from our adult MMA cohort that was characterized by a prevalence towards female patients (78.7%, [34]; 82.5%, [51]), as well as from previous well-established data from the literature [6,9]. Such an issue may be explained by the release of hormones associated with secondary sex characteristics, and it could be a clue to investigate other pathways potentially involved with this obscure disease. As we already reported, MMA incidence shows a bimodal age distribution [9]. Sex hormones could play a particular role in the second peak (45–49 years), which coincides with the age range of physiological menopause in Western countries. However, the specific impact of gender characteristics in MMA has not been fully elucidated so far, especially in pediatric patients. The importance of sex differences in MMA are also reported in a transcriptomic profiling of ICA in adult patients. The study identified 133 and 439 sex-specific differentially expressed genes (DEGs) for men and women through an RNA sequencing (RNAseq) analysis [44]. Another RNAseq study identified a total of 533 DEGs in the peripheral blood of MMA patients, further highlighting the importance of a transcriptomic approach in a multifactorial disease [45].

A considerable amount of evidence in the literature sustained the involvement of cEPCs in MMA etiology and development [33,34,35,36]. However, little is known about their role in the early onset of the disease. By investigating cEPC’s role in a MMA pediatric cohort, the present study showed an increased level of circulating CD45^dim^CD34^+^CD133^+^ mononuclear cells in the peripheral blood of patients. The reported enhanced levels may be associated with a higher cEPC mobilization rate, which is possibly due to the presence of the stenotic lesion leading to the creation of a peculiar network of collateral vessels. Likely, the sprouting and formation of new vessels required the recruitment of cEPCs that, from bone marrow, migrated to the cerebral district. We did not find a statistically significant difference in the cEPC% between the two subgroups of MMA patients (before and after the neurosurgical intervention), thus suggesting that neurosurgical procedures did not affect the percentage of cEPC. Noticeably, in a homogeneous cohort of non-operated Caucasian adult MMA patients, a marked decrease of cEPCs was evident [34]. Although other groups suggested that cEPCs are reduced in MMA pediatric patients, the unmatching in ages among the analyzed groups could have represented the bias underlying such findings [36]. Specifically, the control group considered was characterized by a median age of 23 years, which is an older group compared to the MMA group with a median age of 7.5 years. Therefore, also considering our findings, it seems that age represents a crucial factor in the evaluation of cEPCs as well as the specific methodological approach used for cEPC identification and quantification. Other circulating vascular progenitor cells are also involved in MMA. As previously reported, fibrocellular thickening and proliferation suggested the involvement of smooth-muscle cells [52,53]. A remarkable increase of circulating CD34^+^ cells, which are associated with neovascularization that follows ischemic stress, was observed in MMA patients. Similarly, an increased level of circulating CD34^+^CXCR4^+^ cells in the peripheral blood of MMA patients has been found [54,55].

In addition to the evaluation of circulating progenitor cells, the analysis of putative molecular biomarkers in our series appears to reflect the pathophysiological mechanism of MMA onset. The analysis of CSF samples from our patient cohort showed increased CSF levels of ANG-2 and VEGF-A, possibly reflecting the deregulated angiogenesis typical of the disease. In fact, both VEGF-A and ANG-2 are well-known angiogenic factors [37,38,39]. Indeed, the main MMA pathological feature is the development of fragile compensatory vessels to counterbalance the lack of oxygen and glucose caused by the progressive thickening of the vascular wall of involved arteries of the Willis circle. The increase in VEGF-A found in the CSF of MMA patients may indicate the attempt of enhancing compensatory angiogenesis, though often ineffective. The main clinical manifestations both in our series and in the literature of MMA pediatric patients are ischemic strokes or TIA, which are related to the reduced blood flow not being able to satisfy metabolic requests. Thereby, the clinical onset in children is often induced by hyperventilation subsequent to fatigue or crying [56]. Hemorrhagic stroke is extremely rare in MMA-affected children, with a rate around 2–3%, probably because stenotic and thrombotic mechanisms overbalance the frailty of pathological vessels that are not yet completely developed in the pediatric population [57]. Thus, overexpression of VEGF-A could represent the effect of angiogenic stimuli in the injured brain district. ANG-2 is a proangiogenic factor involved in the formation of new vessels, but it also promotes pathological angiogenesis, vascular permeability, and inflammation. Indeed, ANG-2 is overexpressed in multiple inflammatory diseases and implicated both in the direct control of inflammation-related signaling pathways and in the recruitment of inflammation cells [38].

We did not find modulation of the above-mentioned protein factors in the plasma of MMA patients, which could be due to the low sensitivity of the traditional standard antibody-based approach (ELISA). Future technological advances that allow a more accurate dosage in peripheral blood as well could refine our results, thus paving the way to obtain reliable and noninvasive plasma biomarkers. To overcome these possible limitations, we also benefited from a digital detection system (SiMoA) based on microscopic beads and captured antibodies, which has been recently employed for ultrasensitive protein determination [58,59]. SiMoA enables the detection of femtomolar amounts of selected proteins in a sample, thus allowing the simultaneous determination of analytes of interest. Specifically, we did not find a statistically significant difference for NfL and GFAP release, neither in plasma nor in CSF, when comparing pediatric MMA patients and control subjects.

Interestingly, NfL is considered a crucial biomarker of hypoxic–ischemic brain and a sensitive indicator of white matter axonal damage injury, predicting survival and neurological outcome after pediatric stroke [60,61]. However, despite the small size, NfL levels in plasma and CSF were not different between cases and controls, suggesting that axonal damage could not be present in pediatric or early onset MMA. Indeed, the absence of statistically considerable differences in NfL CSF-release between patients and controls support the hypothesis that the pediatric MMA cohort may have less neuronal damage compared to adult MMA patients (unpublished results, manuscript submitted). As shown in Figure 4c, two (out of nine) children were characterized by the highest levels of detected NfL compared to the rest of the pediatric cohort. Intriguingly, these patients were the only children who underwent neurosurgical bypass within a few days after an acute ischemic attack. Considering the timing of clinical manifestation and symptoms onset, all other children were later referred to our tertiary national institution from other centers. This finding was also related to the rarity of disease in Western countries and to the possible delayed diagnosis.

In children with white matter disorders, elevated CSF GFAP levels, which are suggestive of astrocyte activation response (“reactive astrogliosis”) to brain injury or neuro-perturbative conditions, have been reported [61]. Evidence in the literature showed that astrocyte activation and GFAP-release might be beneficial to the brain recovery process, whereas excessive gliosis and associated neuroinflammatory responses could have a negative impact on the brain’s structural and functional recovery. The present study has not highlighted any relevant difference in GFAP levels, neither in plasma nor in CSF. However, overall, our findings may be influenced by the small size of this pediatric cohort involved, which was due to the rarity of the disease in Western countries. Future investigations on larger pediatric populations are necessary to confirm our results and could help in prospectively analyzing if not only EPCs, but also ANG-2, VEGF-A, NfL, and GFAP CSF levels could estimate and track the progression of recovery after revascularization surgery, and if there are medium- and long-term differences after direct bypass, indirect revascularization procedures, or a combination of both techniques.

## 4. Materials and Methods

### 4.1. Moyamoya Patients and Unrelated Controls: Inclusion Criteria

This was an observational study conducted on MMA patients belonging to the GEN-O-MA study [50]. MMA was diagnosed according to established literature criteria: (i) stenosis or occlusion at the terminal portion of the ICA and/or the proximal portion of the anterior and/or middle cerebral arteries (MCA), (ii) abnormal vascular networks in the vicinity of the stenotic lesions, and (iii) a bilateral presence of these findings [62]. The study was designed as a multicenter (prospective and retrospective) observational cohort study across Italy. The project was coordinated by the Neurology IX—Cerebrovascular Unit of the Fondazione IRCCS Istituto Neurologico Carlo Besta, Milan, Italy. A strong collaborative relationship with the Neuroradiological, Neurosurgical, and Child Neuropsychiatric Unit and Developmental Neurology Division of the same hospital and with other participating centers allowed multidisciplinary collaboration, thus leading to an improved resulting achievement. From the original population of 150 patients consecutively enrolled between November 2014 and October 2022, only pediatric patients (age < 18 years) were selected for the present study (see Table 1 for clinical–demographic characteristics). Twelve out of sixteen patients underwent MCA-STA bypass. Individuals fasted within 12 h, and subjects with endometriosis and/or were positive for HIV, HBV, or HCV were excluded from this study. A group of age- and sex-matched unrelated (UNR) patients were recruited as controls.

### 4.2. Ethical Issues

The study design was approved by the local Ethics Committee and was performed in accordance with the 2013 WMA Declaration of Helsinki. Patients underwent diagnostic procedures and received therapy according to local practice after informed written consent for study participation and sample collection. Privacy procedures were applied to protect patients’ and the healthy control group’s personal identities.

### 4.3. Peripheral Blood, Plasma and CSF Sample Collection

CSF was collected from MA patients and UNR subjects either during neurosurgical interventions or by lumbar puncture. Lumbar puncture and CSF handling after withdrawal followed a structured protocol [63]. After overnight fasting, a lumbar puncture (LP) was performed in the morning by a trained neurologist. CSF (10–20 mL) was collected via gravity grip using a 22-gauge Sprotte spinal needle (Geisingen, Germany). All samples were free of visible blood contamination. After collection, samples were briefly centrifuged (2000× *g* for 15 min) to remove any cellular debris and transferred to another polypropylene sterile tube. CSF was aliquoted (500 uL) into polypropylene tubes and stored at −80 °C for subsequent molecular analyses. Twenty-four milliliters of peripheral blood were withdrawn by venipuncture from MA and UNR subjects and collected in tubes containing ethylenediaminetetraacetic acid (EDTA) as an anticoagulant (Vacuette^®^, Preanalitica s.r.l., Caravaggio, Italy). One vacutainer was stored at −20 °C for future molecular analysis. For plasma collection, two vacutainers were centrifuged for 10 min at 300× *g*; the plasma was then transferred into a new tube (SARSTEDT AG and Co, Nümbrecht, Germany) and stored in aliquots at −80 °C until use.

### 4.4. Clinical–Radiological Factors

For all patients, demographic and clinical features were collected with a standardized form [50].

MA was classified into the bilateral or unilateral types depending on the number of distal ICAs involved, as observed on conventional angiography [62]. The diagnoses of ischemic or hemorrhagic stroke were confirmed with conventional neuroimaging (computerized tomography scan and magnetic resonance imaging). MA severity was assessed with the Suzuki scale [64].

### 4.5. Flow Cytometry Analysis

Flow cytometry analysis of cEPCs was performed on fresh whole blood using Flow-Count Fluorospheres (Beckman Coulter s.r.l., Cassina De’ Pecchi, Italy). Fifty microliters of whole blood (EDTA) mixed with equal volume of Flow-Count Fluorospheres (Beckman Coulter s.r.l., Brea, CA, USA) were incubated with 10 µL of monoclonal antibodies anti-CD34-FITC, 1 µL of anti-CD45-Pe vio770, and 1.5 µL anti-CD133-PE (Miltenyi Biotech, Bergisch Gladbach, Germany) for 30 min at +4 °C in the dark. Flow-Count Fluorospheres are a suspension of fluorescent microbeads used to determine absolute counts on the flow cytometer. Each fluorosphere contains a dye which has a fluorescent emission range of 525 nm to 700 nm when excited at 488 nm. They have uniform size and fluorescence intensity and an assayed concentration, allowing a direct determination of absolute counts. They have been used as an aid in optimizing a quantitative fluorescence-activated cell sorting (FACS) analysis. Then, erythrocytes were lysed and leukocytes were fixed with a Uti-Lyse kit (DakoCytomation, Glostrup, Denmark). A specific staining was determined with the appropriate Isotype Control (BD Bioscience, San Jose, CA, USA), and samples were analyzed within a week of blood collection in a FACSCalibur flow cytometer (BD Bioscience, San Jose, CA, USA) equipped with CellQuest software. We considered cEPCs CD45^dim^CD34^+^CD133^+^ to be mononuclear cells. The cellular population selection criteria allowed us to fine-tune the setting of the flow cytometry parameters by creating gates to distinguish the EPC cellular component of interest in the control group [34]. Because cEPCs are rare in normal peripheral blood, at least 500 CD34+ cells per sample were acquired, and nonviable cells were excluded with physical gating. For sample normalization, a complete white blood cell count was performed with the cell counter Advia 120 (Bayer, Leverkusen, Germany). The percentage of cEPCs was calculated as follows: % cEPCs = (cEPCs/µL/WBC/µL) × 100.

### 4.6. ELISA

Angiopoietin-2 (ANG-2; Boster Biological Technology Co., LTD, Pleasanton, CA, USA), vascular endothelial growth factor-A (VEGF-A; Boster Biological Technology Co., LTD, Pleasanton, CA, USA), metalloproteinase-9 (MMP-9; ThermoFisher, Monza, Italy), and interleukin 6 (IL-6; ThermoFisher) concentrations were assessed using a highly sensitive ELISA kit in triplicate on the plasma and CSF samples. The assays were conducted in a 96-well plate, and all working solutions were prepared according to the manufacturer’s instructions. Blank, standards, and plasma and CSF controls and samples in triplicate were included in each plate. Briefly, 100 μL of plasma and CSF samples were added to each well, and the plate was then covered and left at 37 °C for 90 min. Then, the plate was washed three times with a wash buffer, 100 μL of Biotinylated Antibody Reagent was added into each well, and the plate was then incubated at 37 °C for 1 h. After that, three washings were carried out, and 100 μL of the streptavidin-HRP solution was added to each well. The plate was covered and incubated for 30 min at 37 °C or room temperature. After a final wash step, 90 or 100 μL of color-developing reagent were added to each well, and the plate was developed in the dark for 15–20 min at 37 °C. 100 μL of stop solution were added to each well to stop the reaction, and the absorbance was measured on a plate reader at 450 nm. The average of three determinations (ng/mL or pg/mL) was used for statistical analysis.

### 4.7. SIMOA Ultra-Sensitive Digital Immunoassay for NF-Light and GFAP (Single Molecule Array SiMoA™)

Quantitative determination of NF-light (NfL) and glial fibrillary acidic protein (GFAP) in plasma and CSF samples from UNR subjects and MA pediatric patients was performed with single molecule array (SiMoA) technology on the SR-X analyzer from Quanterix (Billerica, MA, USA) according to the manufacturer’s instructions. NfL and GFAP concentrations were determined using the commercially available kit SiMoA Neuro 2 Plex B Advantage Kit (Item 103520, Quanterix). Reagents, such as bead reagent, detector reagent, SBG (Streptavidin-b-galactosidase) reagent, RGP (Resorufin b-D-galactopyranoside) reagent, and SR-X sample diluent as well as reference calibrators (n = 8), controls (n = 2), and samples have been equilibrated to room temperature for 1 h prior to the immunoassay procedure. Briefly, plasma and CSF samples were thawed at room temperature, centrifuged at 10,000× *g* for 5 min, and then diluted with the provided dilution buffer according to the manufacturer’s protocols. Calibrators and two quality controls of known concentrations were run in duplicate. The average of the two determinations (pg/mL) was used for statistical analysis. Samples with coefficients of variation (CV) higher than 20% were rerun with appropriate calibrators and controls. Mean intra-assay CVs were between 0.1% and 19%.

### 4.8. Statistics and Data Visualization

Data were expressed as mean ± standard deviation (SD), and statistical significance (* *p*  <  0.05; ** *p* < 0.01) was calculated with Student’s *t*-test by using GraphPad Prism 8 software (GraphPad Software, Inc., San Diego, CA, USA). Differences were considered to be significant at *p*  <  0.05. The values of at least three independent experiments are shown. For each of the investigated proteins, numerically homogeneous groups of MA patients and UNR subjects were compared. Distribution normality was examined with Shapiro-Wilk’s test and, if data were normally distributed, for multiple group comparisons.

## 5. Conclusions

Pediatric MMA patients showed some peculiar cellular and molecular characteristics compared to age- and sex-matched control subjects, both in blood and CSF. Moreover, we highlighted differences in cEPC% between children and adult patients with MMA. Despite the limited cohort of patients, our findings could suggest some fields of investigation to be confirmed by wider, perspective studies, with the final aim to identify predictive and prognostic circulating biomarkers and potential therapeutic targets for personalized care of MMA pediatric patients.

## Figures and Tables

**Figure 1 ijms-24-01233-f001:**
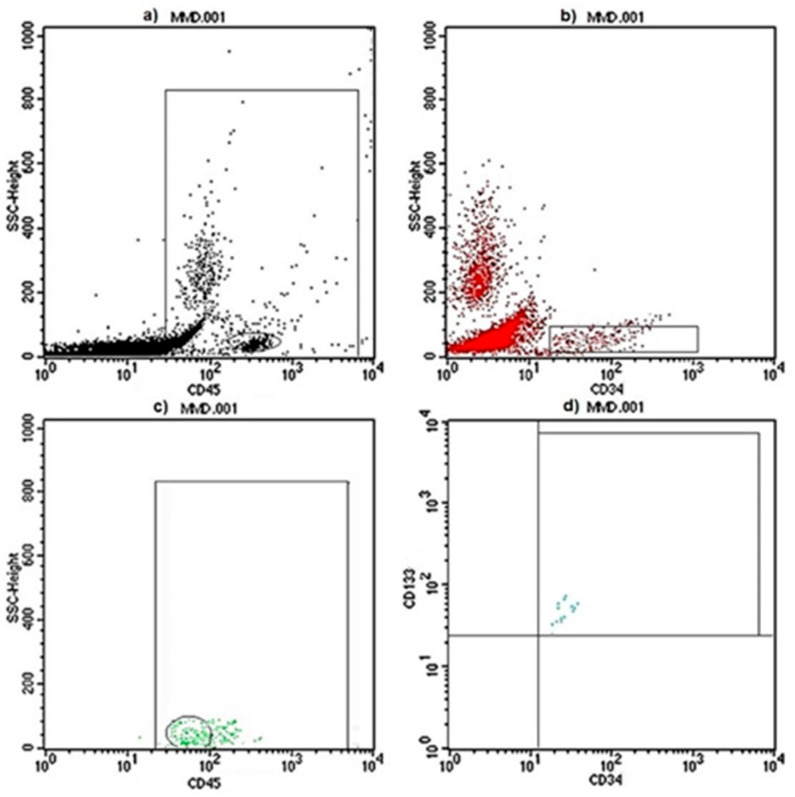
Flow cytometer analysis of circulating EPCs. The EPC counts in the peripheral blood were determined with flow cytometry on whole blood samples: (**a**) CD45^+^ events on Gate1 region using PE-Vio770-labeled antibodies against CD45; (**b**) proportion of CD34^+^ cells on CD45^+^-gated events as analyzed in the Gate2 region using FITC-labeled antibodies against CD34; (**c**) Gate3 region-selected CD45^dim^ cells; (**d**) triple-positive cells were identified by the dual expression of CD34 and CD133 (PE-labeled antibodies) within the CD45^dim^-gated population. Because EPCs are rare in normal peripheral blood, at least 500 CD34^+^ events (**b**) per sample were acquired.

**Figure 2 ijms-24-01233-f002:**
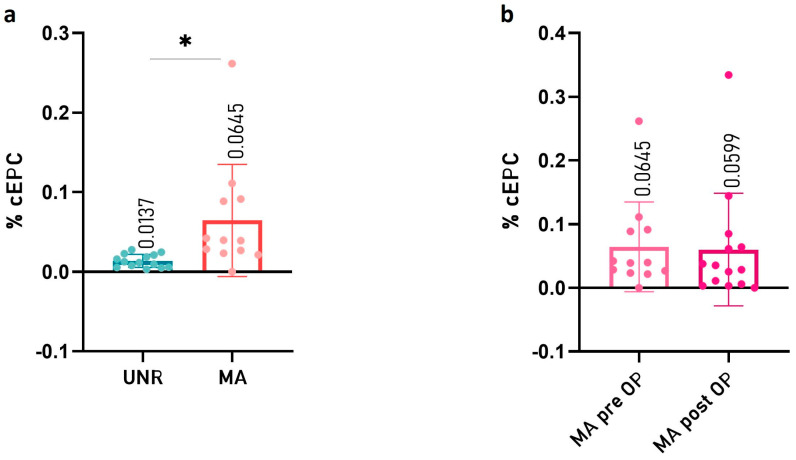
Circulating endothelial progenitor cell (cEPC) in whole blood. (**a**) cEPC levels from a preoperatory set of MMA patients (n = 12) as compared with unrelated controls (UNR, n = 14). (**b**) cEPC levels from a homogeneous group of MMA patients before (pre OP, n = 12) and after (post OP, n = 14) the neurosurgical procedures. Data are expressed as the mean of cEPC% ± SD, where the cEPC% value was calculated as follows: (cEPCs/µL/White Blood Cells/μL) × 100. Error bars represent mean ± SD. The statistical significance (* *p* < 0.05) was calculated using two-tailed Student’s *t*-test (*p* = 0.0129).

**Figure 3 ijms-24-01233-f003:**
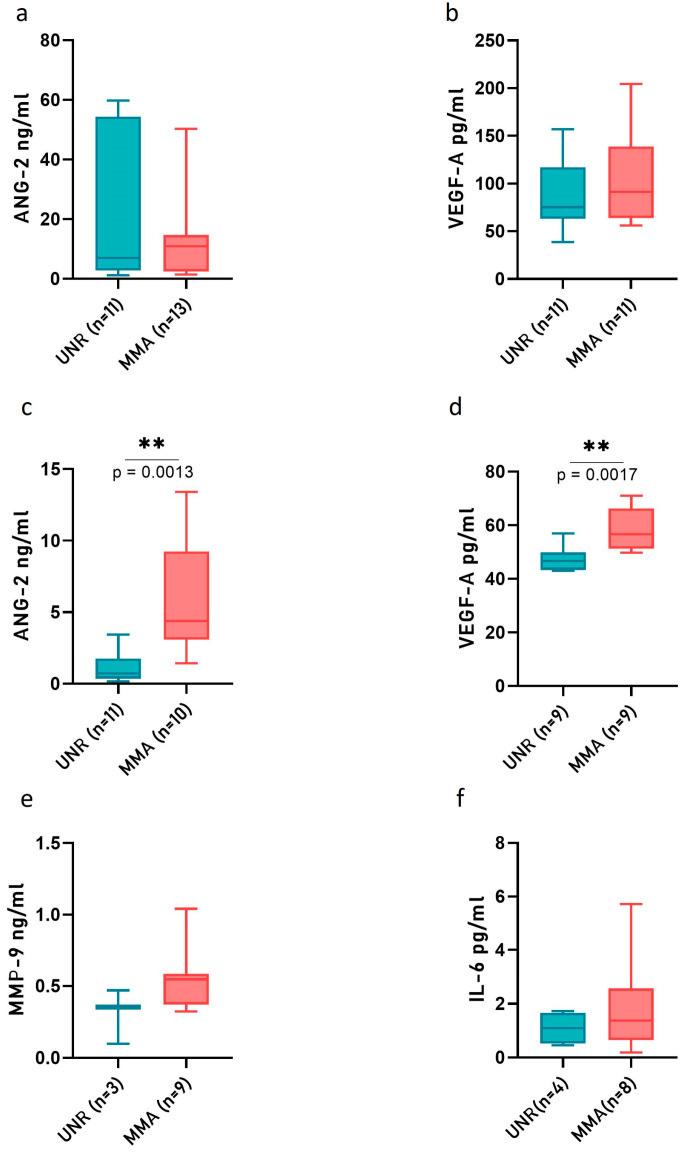
Plasma and CSF concentrations of angiogenic/inflammatory factors. The plasma (**a**,**b**) and CSF (**c**–**f**) levels of angiopoietin-2 (Ang-2), vascular endothelial growth factor-A (VEGF-A), matrix metalloproteinase (MMP-9) and interleukin-6 (IL-6) were evaluated with ELISA in MMA patients and UNR subjects. The boxes represent data obtained in the range of the 25th–75th percentile; the line across the boxes indicates the median value; the lines above and below the boxes indicate extreme values (10th or 90th percentile). The statistical significance (** *p* < 0.01) was calculated through Student’s *t*-test. The values of at least three independent experiments are shown.

**Figure 4 ijms-24-01233-f004:**
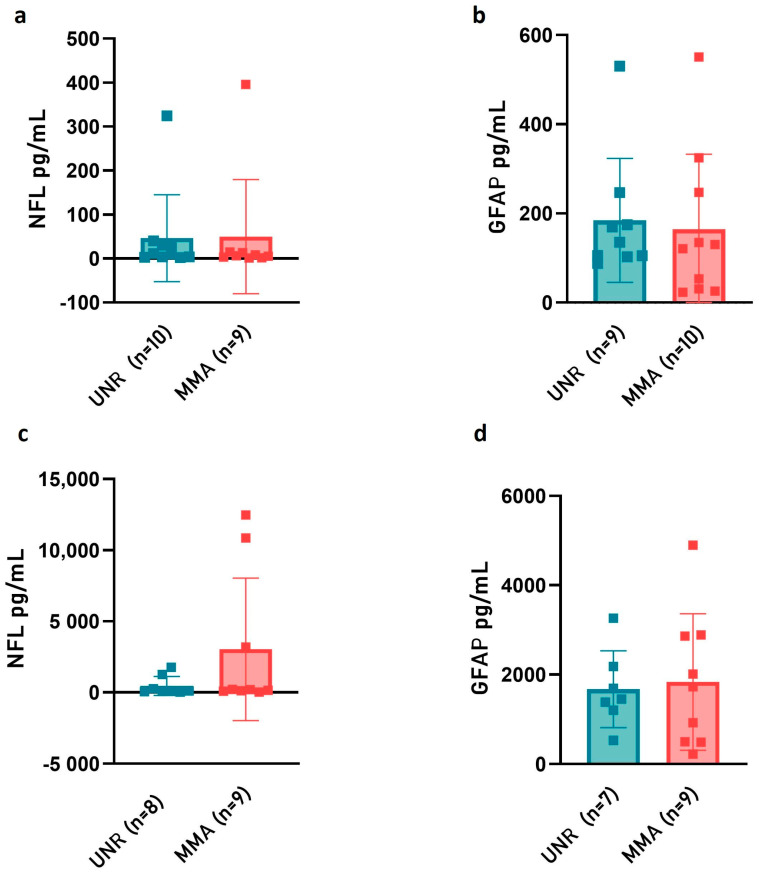
Plasma and CSF concentrations of structural protein factors. The plasma (**a**,**b**) and CSF (**c**,**d**) levels of neurofilament light (NfL) and glial fibrillary acidic protein (GFAP) proteins were determined with SiMoA in MMA patients and UNR subjects. Error bars represent mean ± SD. The statistical significance was calculated through Student’s *t*-test. The values of at least three independent experiments are shown.

**Table 1 ijms-24-01233-t001:** Demographic, clinical, and neuroradiological features of MMA patients whose samples were included in biological analyses.

ID	Age (y)	Gender	CVD Type	NIHSS	U-B	Suzuki Grading	Pharmacological Therapy	Indirect R	Direct R	Genetic Syndromes
MMA8	16	F	none	0	B	III	AG, ASA	yes	yes	no
MMA12	16	F	IS	1	B	III	/	no	yes	no
MMA14	15	F	TIA	4	B	IV	AG, ASA	yes	yes	no
MMA16	11	M	IS	6	U	III	AG, ASA, AE	no	no	no
MMA18	9	M	TIA	0	B	IV	AG, ASA, AE	no	no	no
MMA22	5	M	IS	2	B	III	AG, ASA, AE	yes	yes	no
MMA44	9	M	TIA	0	B	IV	AG, ASA	yes	yes	no
MMA46	6	F	TIA	0	B	IV	ASA	yes	yes	u/i
MMA57	11	M	none	0	U	III	AG, ASA	yes	yes	NF-1
MMA60	9	M	IS	2	B	IV	/	yes	yes	no
MMA62	7	F	TIA	0	B	IV	AG, ASA	yes	yes	no
MMA64	3	M	IS	0	B	III	AG, ASA, AE	yes	yes	NF-1
MMA65	7	F	IS	1	B	IV	AG, ASA	yes	yes	no
MMA67	6	M	IS	1	B	IV	AG, ASA	yes	yes	no
MMA73	9	F	none	0	B	III	AG, ASA	no	no	DS
MMA77	8	F	none	0	B	IV	AG, ASA, AE	no	no	DT4q

AE, antiepileptic; AG, antiaggregants; ASA, acetylsalicylic acid; B, bilateral; CVD, cerebrovascular disease; DS, Down syndrome; DT4q, distal trisomy 4q; F, female; IS, ischemic stroke; M, male; MMA, moyamoya arteriopathy; NIHSS, National Institute of Health scale; NF-1, neurofibromatosis type 1; TIA, transient ischemic attack; u/i, under investigation; U, unilateral; R, neurosurgical revascularization; y, years.

## Data Availability

Data supporting the reported results can be found in publicly archived datasets generated during the study at the Fondazione IRCCS Istituto Neurologico Carlo Besta (https://zenodo.org/communities/besta/?page=1&size=20; accessed on 15 November 2022).

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
