# Peer review of "Increase of Circulating Endothelial Progenitor Cells and Released Angiogenic Factors in Children with Moyamoya Arteriopathy"

_ijms, 2023, doi:10.3390/ijms24021233_

Round 1
Reviewer 1 Report
The manuscript presented from Gorla et al., entitled "Circulating Biomarkers in Children with Moyamoya Arteriopathy" is interesting and original. However the authors limited their experimental research only to ELISA. Recent studies reported other interesting approach as RNA-seq in patients with this disease. But the authors did not mention or discussed it.
The conclusions does not reflect the results because the authors does not find new target as they describe in the text.
At the same time they did not explain or show the mechanism of the pathology. To improve the quality of this article they should perform RNA-seq or analyzed previous dataset from patients with MAM.
The authors should explain how they selected the data reported in graph. SEM?
FACS plots and sorting strategy should be also showed as main figure.
Xu S, Wei W, Zhang F, Chen T, Dong L, Shi J, Wu X, Zhang T, Li Z, Zhang J, Li X and Chen J (2022) Transcriptomic Profiling of Intracranial Arteries in Adult Patients With Moyamoya Disease Reveals Novel Insights Into Its Pathogenesis. Front. Mol. Neurosci. 15:881954. doi: 10.3389/fnmol.2022.881954
Author Response
Dear Editors,
Dear Dr Echo Liu
(Section Manager Editor),
Milano, December 21st 2022
MANUSCRIPT ID ijms-2096975: Circulating Biomarkers in Children with Moyamoya Arteriopathy
Dear Editors,
Please find enclosed the revised version of the manuscript entitled “Increase of Circulating Endothelial Progenitor Cells and Released Angiogenic Factors in Children with Moyamoya Arteriopathy” that we resubmit to Editors on behalf of all the authors, after revising and editing the text (marked up using the “Track Changes” function), thanks to Editor and Reviewer’s suggestions. Detailed answers to Editors and Reviewers are reported below.
We are confident that the new information and additional data that we have included will be convincing for the Editor and Reviewers and will fill the gaps highlighted by the Referees.
Thereby, we hope that our paper could now be suitable for publication in the section “Molecular Neurobiology”, Topical Collection “Featured Papers-Molecular Neurobiology”, in International Journal of Molecular Sciences.
Thank you for your consideration.
Your sincerely,
Laura Gatti
Dr. Laura Gatti, PhD
Head of Cellular Neurobiology Laboratory
Neurology IX Cerebrovascular Disease Unit
Via Celoria 11, 20133
Fondazione IRCCS Istituto Neurologico Carlo Besta, Milan, Italy
laura.gatti@istituto-besta.it
Comments and Suggestions for Authors
Reviewer 1
- The manuscript presented from Gorla et al., entitled "Circulating Biomarkers in Children with Moyamoya Arteriopathy" is interesting and original. However, the authors limited their experimental research only to ELISA.
We are grateful to the Reviewer for his/her comments and for the careful revision. We admit that our paper is limited by a small sample size of pediatric patients and shows few positive results. However, in addition to ELISA results, we have included circulating EPCs evaluation in peripheral blood of Moyamoya artheriopathy (MMA) children and control subjects, and ultra-sensitive/digital SiMoA analyses. This piece of work is novel and original, in consideration of the heterogeneity of previous published results, obtained with different applied methodologies, also reminding the rareness of this cerebrovascular disease. We believe that our study may be helpful to clarify the significance of cEPCs and of other selected circulating protein factors in pediatric MMA. Moreover, we are confident that the new information and data that we have included in the revised version, will be convincing and will fill the gaps highlighted by the Referee.
- Recent studies reported other interesting approach as RNA-seq in patients with this disease (Xu S, Wei W, Zhang F, Chen T, Dong L, Shi J, Wu X, Zhang T, Li Z, Zhang J, Li X and Chen J (2022) Transcriptomic Profiling of Intracranial Arteries in Adult Patients With Moyamoya Disease Reveals Novel Insights Into Its Pathogenesis. Front. Mol. Neurosci. 15:881954. doi: 10.3389/fnmol.2022.881954). But the authors did not mention or discussed it. The conclusions does not reflect the results because the authors does not find new target as they describe in the text. At the same time they did not explain or show the mechanism of the pathology. To improve the quality of this article they should perform RNA-seq or analyzed previous dataset from patients with MAM.
We greatly appreciated the comment of the Reviewer. Unfortunately, we dealt with the unavailability of the stenotic brain arterial (MCA) fragments in pediatric patients and –even more- in pediatric control subjects. Moreover, due to the long experimental time requested to perform RNAseq on RNA samples extracted by the rare circulating progenitors in plasma/CSF, and considering the short time for the revision, we overcome such obstacle by providing results obtained in adult patients by other Authors. Specifically, this part has been included in Discussion, on page 9, lines 16-21 and the suggested references have been included in the Bibliography (ref n°44, 45 respectively).
- The authors should explain how they selected the data reported in graph. SEM?
We thank the Referee for this comment. The average of two determinations (ng/ml or pg/ml) was used for statistical analysis. Samples with coefficients of variation (CV) higher than 20% were re-run with appropriate calibrators and controls. Mean intra-assay CVs were between 0.1% and 19%. We added details on data selection and statistical analyses in Methods (Paragraph 4.8. Statistics and Data Visualization, on page 13, lines 20-26) and in legends to Figures 2 (previous Figure 1) and Figure 4 (previous Figure 3), on page 6, lines 6-7 and page 8, line 10, respectively.
- FACS plots and sorting strategy should be also showed as main figure.
We thank the Referee for this comment. It reminds to the more general and still open question about the MMA patient selection criteria and the choice of the best cytofluorimetric markers useful to characterize the EPC population. Many previous studies have studied cEPCs to better understand and characterize the disease pathogenesis. However, the controversial results about EPC in MMA speculate about the involvement of such cells in this rare condition and about the heterogeneity of the used patient samples. Noticeably, differences between studies in the % of EPC of MMA patient could be due to differences in methodological approach and in the selection of the patient cohort. According to the Reviewer suggestion, we now display FACS analysis plots (see below), to point out the applied cell population selection criteria, which based on two previous publications in the field by our Institution [(i) Corsini, E. et al, J. Neurooncol. 2012, 108, 123–129 doi: 10.1007/s11060-012-0805-8; (ii) Tinelli, F et al. Int J Mol Sci, 2020, 21, 5763, that is current ref 34]. Our previous papers allowed us to fine-tune the setting of the flow cytometer parameters by creating gates to distinguish the EPC cellular component of interest, in a control group of healthy donors firstly. Based on this setting, it was then possible to conduct subsequent analyses on peripheral blood from MMA patients and UNR subjects. We have included such information in Results as Figure 1 (page 5, lines 6-14). Therefore, we have re-numbered all the successive Figures (from Figure 2 to Figure 4).
Figure 1. Flow cytometer analysis of circulating EPCs. The EPC counts in the peripheral blood were determined by flow cytometry on whole blood samples: (a) CD45+ events on Gate1 region using VIO PRCP700-labeled antibodies against CD45; (b) Proportion of CD34+ cells on CD45+-gated events was analyzed in the Gate2 region using FITC-labeled antibodies against CD34; (c) Gate3 region selected CD45dim cells; (d) Triple-positive cells were identified by the dual expression of CD34 and CD133 within the CD45dim-gated population. Since EPCs are rare in normal peripheral blood, at least 500 CD34+ events (b) per sample were acquired.

Reviewer 2 Report
In the manuscript presented by Gemma Gorla et al., the authors included 16 MMA pediatric patients and 14 unrelated pediatric subjects to identify predictive/prognostic circulating biomarkers. The levels of some angiogenic/inflammatory factors and structural proteins in plasma and CSF were measured. In addition, the cEPC% level was compared between the two groups. The authors concluded that pediatric MMA patients showed some peculiar cellular and molecular characteristics compared to age/sex matched control subjects. The findings are encouraging and of some significance, but there are a number of issues that need to be addressed.
1. Only a few angiogenic/inflammatory factors as well as cEPC were selected in this study as potential biomarkers, which is not enough to aim the purpose of assessing circulating biomarkers in children with moyamoya arteriopathy. The title needs to be modified as well as the purpose of this study in the Abstract and Introduction sections.
2. The angiogenic/inflammatory factors need to be described in the Introduction section. There are some other cell types in the circulation besides cEPCs. Why cEPCs are solely important in MMA need to be stated in the Introduction or Discussion section.
3. Why did the authors select Ang-2, VEGF-A, MMP-9 and IL-6 as potential angiogenic/inflammatory markers?
4. In section 2.1, “…unrelated controls (UNR), including subjects suffering from different diseases not closely related to MMA (e.g., epilepsy,…”. As far as I know, MMA can also cause seizures, so MMA is not unrelated to epilepsy.
5. Why were MMP-9 and IL-6 only measured in CSF but not in the plasma? Please explain the difference in the number of patients between different tests.
6. The purpose of measuring the levels of NfL and GFAP needs to be stated in the beginning of 2.4.
7. In the figure legend of Figure 3, the “plasma (a) and CSF (b)” needs to be corrected to be in consistent with the figure.
8. In the Materials and Methods section, the diagnosis criteria in the literature needs to be described as well as the methodology of this study, even though the references were listed.
Author Response
Dear Editors,
Dear Dr Echo Liu
(Section Manager Editor),
Milano, December 21st 2022
MANUSCRIPT ID ijms-2096975: Circulating Biomarkers in Children with Moyamoya Arteriopathy
Dear Editors,
Please find enclosed the revised version of the manuscript entitled “Increase of Circulating Endothelial Progenitor Cells and Released Angiogenic Factors in Children with Moyamoya Arteriopathy” that we resubmit to Editors on behalf of all the authors, after revising and editing the text (marked up using the “Track Changes” function), thanks to Editor and Reviewer’s suggestions. Detailed answers to Editors and Reviewers are reported below.
We are confident that the new information and additional data that we have included will be convincing for the Editor and Reviewers and will fill the gaps highlighted by the Referees.
Thereby, we hope that our paper could now be suitable for publication in the section “Molecular Neurobiology”, Topical Collection “Featured Papers-Molecular Neurobiology”, in International Journal of Molecular Sciences.
Thank you for your consideration.
Your sincerely,
Laura Gatti
Dr. Laura Gatti, PhD
Head of Cellular Neurobiology Laboratory
Neurology IX Cerebrovascular Disease Unit
Via Celoria 11, 20133
Fondazione IRCCS Istituto Neurologico Carlo Besta, Milan, Italy
laura.gatti@istituto-besta.it
Comments and Suggestions for Authors
Reviewer 2:
- In the manuscript presented by Gemma Gorla et al., the authors included 16 MMA pediatric patients and 14 unrelated pediatric subjects to identify predictive/prognostic circulating biomarkers. The levels of some angiogenic/inflammatory factors and structural proteins in plasma and CSF were measured. In addition, the cEPC% level was compared between the two groups. The authors concluded that pediatric MMA patients showed some peculiar cellular and molecular characteristics compared to age/sex matched control subjects. The findings are encouraging and of some significance, but there are a number of issues that need to be addressed.
We are grateful to the Reviewer for his/her comments and for the careful revision. We are confident that the new information and additional data that we have included in the revision version, will be convincing and will fill the gaps highlighted by the Referee.
- Only a few angiogenic/inflammatory factors as well as cEPC were selected in this study as potential biomarkers, which is not enough to aim the purpose of assessing circulating biomarkers in children with moyamoya arteriopathy. The title needs to be modified as well as the purpose of this study in the Abstract and Introduction sections.
We thank the Referee for this comment. We have modified accordingly the manuscript Title (page 1) and also the purpose of this study, both in the Abstract (page 1, lines 41-42) and Introduction (page 4, lines 1-5) sections.
The new title is: “Increase of Circulating Endothelial Progenitor Cells and Released Angiogenic Factors in Children with Moyamoya Arteriopathy”
- The angiogenic/inflammatory factors need to be described in the Introduction section. There are some other cell types in the circulation besides cEPCs. Why cEPCs are solely important in MMA need to be stated in the Introduction or Discussion section.
We greatly appreciated the comment of the Reviewer. We have now described the angiogenic/inflammatory factors in the Introduction section (page 3, lines 24-48) with particular reference to results already obtained in adult MMA cohorts. Moreover, we have included details on some other cell types in the circulation besides cEPCs and we have stated the relevance of cEPCs in MMA, in Discussion (page 9, lines 41-47). We have accordingly modified the Bibliography, by including refs n° 37-49 and 52-55.
- Why did the authors select Ang-2, VEGF-A, MMP-9 and IL-6 as potential angiogenic/inflammatory markers?
We have previously performed a molecular profiling of several potential angiogenic/inflammatory markers, both in plasma (Dei Cas et al 2021) and CSF (Potenza et al, manuscript under revision) of adult MMA patients, and we have obtained promising results from assaying the release of such selected proteins. We now aimed to characterize the molecular profile also in pediatric patients, based on these previous results and on literature data. We are confident that the selection of Ang-2, VEGF, MMP-9 and IL-6 could results clearer thanks to the addition of information in the Introduction (page 3, lines 24-48) section (see above).
- In section 2.1, “…unrelated controls (UNR), including subjects suffering from different diseases not closely related to MMA (e.g., epilepsy,…). As far as I know, MMA can also cause seizures, so MMA is not unrelated to epilepsy.
Thank you for your clarification. As you correctly noted, seizures are associated to MMA. Recently, Lu et al reported 15.5% of affected MMA children with a history of seizure [Lu J et al. Electroencephalographic features in pediatric patients with moyamoya disease in China. Chin Neurosurg J. 2020 Jan 13;6:3. doi: 10.1186/s41016-019-0179-2], and also in other pediatric series the rate of epilepsy related to MMA varies between 11 and 18% [Nakase H et al. Long-term follow-up study of “epileptic type” moyamoya disease in children. Neurol Med Chir (Tokyo). 1993;33(9):621–624. doi: 10.2176/nmc.33.621; Ma Y et al. Risk factors for epilepsy recurrence after revascularization in pediatric patients with moyamoya disease. J Stroke Cerebrovasc Dis. 2018;27(3):740–746. doi: 10.1016/j.jstrokecerebrovasdis.2017.10.012].
Nevertheless, the common mechanism underlying seizures among MMA children is represented by ischemic stroke, that generated a 1.62-fold risk of seizure compared with those without an ischemic stroke. The UNR controls employed in our manuscript are represented by children with primary epilepsy, in which hemodynamic mechanism have been excluded. Moreover, we have also dealt with the lack of pediatric biological samples, compared to adult or elderly population. We have better specified this aspect in the text (Results section, paragraph 2.1, on page 4, lines 26-27).
- Why were MMP-9 and IL-6 only measured in CSF but not in the plasma? Please explain the difference in the number of patients between different tests.
We have previously evaluated the level of MMP-9 and IL-6 in plasma (Dei Cas et al 2021) and CSF (Potenza et al, manuscript under revision) of adult MMA patients, and we have obtained more reliable data in CSF, probably due to proximity with the injured brain suffering of arterial stenosis. We think that CSF collected without lumbar puncture, during indispensable neurosurgical procedures, may mirror MMA neuropathological changes, representing an ideal source for disease biomarkers. As we stated above, we have also dealt with the lack of pediatric biological samples (both for blood and CSF), as compared to adult or elderly population, due to clear ethical issues and lack of parents’ informed consent.
- The purpose of measuring the levels of NfL and GFAP needs to be stated in the beginning of 2.4.
We appreciated the suggestion of the Reviewer and we modified the beginning of 2.4 paragraph, by inserting the purpose of measuring the levels of NfL and GFAP (page 7, lines 11-12; page 8, lines 1-2).
- In the figure legend of Figure 3, the “plasma (a) and CSF (b)” needs to be corrected to be in consistent with the figure.
We apologize for the mistakes in the previous Figure 3 (current Figure 4) legend. We have now corrected it, accordingly to the figure contents and to Reviewer suggestions. Specifically: plasma (a, b) and CSF (c, d) (page 8, lines 8-10).
- In the Materials and Methods section, the diagnosis criteria in the literature needs to be described as well as the methodology of this study, even though the references were listed.
We greatly appreciated the comment of the Reviewer, thus we have extensively described in Material and Methods the diagnosis criteria in the literature (page 11, lines 6-10) and the full methodology of our study (page 11, lines 10-17).

Round 2
Reviewer 1 Report
The authors satisfied all my concerns
Reviewer 2 Report
The authors have addressed all my concerns.